## RESEARCH ARTICLE

# Unusual outcome variances as a method to identify potentially problematic clinical trials

**Philippe P. Hujoel** [1,2]*, **Margaux L.A. Hujoel** [3,4]

**1** Department of Epidemiology, School of Public Health, University of Washington, Seattle, Washington, United States of America, **2** Department of Oral Health Sciences, School of Dentistry, University of Washington, Seattle, Washington, United States of America, **3** Department of Human Genetics, University of California, Los Angeles, California, United States of America, **4** Department of Computational Medicine, University of California, Los Angeles, California, United States of America

* hujoel@uw.edu

## Abstract

An unusual outcome variance contributed to uncovering major cases of research misconduct, leading to over 200 retractions. Detecting such problematic randomized trials early – before they unduly influence clinical guidelines – remains challenging. Empirical evidence indicates that differences in variances between trial arms (DiVB-TAs) are usually small and non-significant in properly conducted trials. This study investigated whether the converse – unusually large and statistically significant DiVBTAs - can serve as a red flag for potentially problematic trials. We conducted simulations to assess the sensitivity and specificity of a DiVBTA-based decision rule under realistic scenarios, including proper randomization, heterogeneous treatment effects, and missing-not-at-random data. In parallel, we applied the rule in a real-world analysis of 226 systematically sampled randomized trials in diabetes research to assess whether unusually large and statistically significant DiVBTAs occur with sufficient frequency to warrant screening. Unusually large DiVBTA values were defined as those falling outside the 3-sigma prediction limits. Simulations demonstrated high specificity, with legitimate trials rarely flagged (low false-positive rate), and adequate sensitivity for detecting a specific form of severe fabrication. In the empirical analysis, 19 out of 226 trials (8%) were flagged as potentially problematic demonstrating utility to screening trials for unusually large and statistically significant DiVBTAs. Subsequent screening of the identified trials revealed additional concerns in 18 (out of 19) flagged trials. These findings suggest that screening for unusually large, statistically significant DiVBTAs offers a simple, low-effort tool to identify trials warranting further scrutiny, potentially strengthening the reliability of evidence used in clinical guidelines.

## Introduction

Problematic clinical trials are widespread and erode the credibility of health information. It is estimated that 25% of published clinical trials may be flawed or fraudulent

**Data availability statement:** All relevant data are within the paper and its Supporting Information files. Data, analysis code, and documentation for both the simulations, the case-study, and the Boldt trial presented in the 2nd paragraph of the discussion are available at: https://github.com/mhujoel/DIVBTA.

**Funding:** The author(s) received no specific funding for this work.

**Competing interests:** The authors have declared that no competing interests exist.

[1]. In absolute terms, hundreds of thousands of trials are believed to lack credibility [2]. These unreliable trials have permeated meta-analyses and clinical guidelines [3–6], which has led to concerted efforts at curbing their influence. These efforts have included global regulatory guidelines which have imposed requirements on data collection that are designed to minimize fraud, misrepresentation, and data integrity issues [7].

The conduct of systematic reviews presents itself with its own set of challenges to prevent potentially problematic trials from infiltrating clinical guidelines. A 2021 Cochrane editorial warned of the threat posed by untrustworthy or "problematic" studies, highlighting that retracted studies are only the tip of the iceberg. The Cochrane editorial coincided with the release of new Cochrane guidelines on how to handle concerns about the trustworthiness of a publication when no formal post-publication correction exists [8]. Checklists to identify problematic trials [9–12] and recommendations on how these tools should become integrated during research synthesis followed [13].

The 2021 Cochrane editorial also underscored the urgent need for validated statistical methods to reliably and fairly detect trials with statistical irregularities [8]. One particularly promising method involves identifying improbable distributions of baseline data across trial arms [14]. This method exploits the cornerstone assumption that participants are allocated randomly to interventions leading to predictable distributions of baseline variables across groups. Trials flagged as having highly unusual distributions when compared to those expected by chance have shown a higher likelihood of retraction [15–17]. Additional statistical methods have become available and at least two statistical packages integrate methods to re-appraise the publication integrity in groups of randomized controlled trials [18–21].

An unusual outcome variance – described as a "very small standard deviation" by the first whistleblower– helped in the discovery of the largest fraudulent research body identified to date [22]. Building on the informativeness of unusual standard deviations to inform on fraud, we propose that Differences in Variance Between Trial Arms (DiVBTAs) of continuous outcome measures can offer the basis to develop an objective method to identify potentially problematic trials.

Empirical evidence in support of this proposal is that a preponderance of meta-analyses of DiVBTAs within the setting of clinical trials demonstrated that DiVBTAs are typically not statistically significant, or, when statistically significant, are small in size [23–27]. The power of statistical tests furthermore to detect significant DiVBTA is low in both meta-analyses of clinical trials, and, especially so within the setting of single clinical trials [21].

The discovery of statistically significant DiVBTAs within the setting of a single clinical trial can thus be regarded as a somewhat unexpected finding. Explanations for such unexpected findings currently focus on genuine design and analysis issues such as randomization, heterogeneous treatments effects, informative trial participant dropout, compliance issues, or floor and ceiling effects of the outcome variable. We suggest here that fraud or unintentional error needs to be included in the list of plausible explanations. As such, we (1) describe methods to identify statistically

significant DiVBTA outliers, (2) perform simulations to assess how likely genuine design and analysis issues can cause statistically significant DiVBTA outliers and (3) provide a case-study on clinical trials included in systematic reviews on diabetes.

## Methods

The methods section is presented in two parts (i) simulations to assess the sensitivity and specificity of the proposed DiVBTA decision rule, and (ii) a diabetes case study to assess whether the proposed decision rule has real-world clinical utility. The following background presents DiVBTA terms discussed in the two proceeding subsections.

*Background:* The proposed decision rule to flag a potentially problematic trial has two elements: (1) the DiVBTA has to be statistically significant, and (2) the DiVBTA needs to be an outlier, i.e., fall outside a tolerance band.

A first step is selecting a DiVBTA statistic among those available (for a review of DiVBTA statistics see [21]). For the detection of potentially problematic trials, it is advantageous to focus on those DiVBTA estimators (a) which can be derived from published summary statistics, (b) which are standardized by the mean, and (c) which are normally distributed and robust.

First, selecting a DiVBTA estimator which can be derived from published summary statistics is crucial given that it remains uncommon for authors of clinical trials to share individual participant data. A 2019 survey of authors from 619 randomized controlled trials published in high-impact anesthesiology journals (2014–2016) found that only about 4% provided individual participant data upon request [28]. A 2019 randomized controlled trial assessing the impact of financial incentives to encourage data sharing reported that none of the investigators provided individual participant data [29]. As a result, DiVBTA estimators requiring individual participant data for calculations remain currently of little value in identifying potentially problematic trials.

Potential summary statistics of interest to calculate DiVBTA measures are the standard deviation (s), the mean ($\bar{x}$), and the derived measure of coefficient of variation (CV). For a two-arm trial, $CV_T = \frac{s_T}{\bar{x}_T}$ and $CV_C = \frac{s_C}{\bar{x}_C}$ where subscripts T and C denote treatment and control, respectively.

Second, DiVBTA measures which minimize assumption about mean-variance relationships have been recommended over measures which are built on the assumption that no mean-variance relationships exists [30]. The log coefficient of variation ratio ($ln\left(\frac{CV_T}{CV_C}\right)$ or lnCVR) is from this perspective a conservative choice as it explicitly normalizes variability by the mean. The lnCVR has the other advantage of being a "master" statistic, a statistic which simplifies to other DiVBTA statistics when no mean-variance relationship exists. The log of the variability ratio ($ln\left(\frac{s_T}{s_C}\right)$ or lnVR) is a special DiVBTA case of lnCVR when group means are equal. The F-ratio ($\frac{s_T^2}{s_C^2}$), another DiVBTA ratio measure, is a log transformation of VR (ln(F)=2 lnVR).

And third, normally distributed ratio DiVBTA measures are preferable when it comes identifying outliers. A log transformation can achieve this goal by reducing skewness which is, for instance, inherent to the F-statistic. Ratio DiVBTA measures are preferable to difference DiVBTA measures because they are scale-invariant, robust across heterogeneous studies, and largely unaffected by errors in publications that mislabel standard errors as standard deviations or fail to label the reported measures of variability as either standard errors or standard deviations.

(i) The first criterion needed for a DiVBTA-based decision rule is to establish statistical significance of the DiVBTA. Methods to test the statistical significance of lnCVR are presented in the section on simulation methods and in the Supplementary Materials for lnVR and the F statistic (S1 Text).

(ii) The second criterion needed for a DiVBTA-based decision rule is to define an outlier, i.e., to construct a DiVBTA tolerance band. Standard trial dynamics (which can lead to statistically significant DiVBTAs) should not be flagged as potentially problematic. Randomization, for instance, will in and of itself lead to 5% of the DiVBTAs to be statistically significant when the type I error rate is set at 5%. Thus, 5% of the trials would be falsely flagged as potentially

problematic without setting a DiVBTA tolerance band. Other legitimate trial dynamics such as heterogeneous treatment effects may further increase the proportion of falsely flagged trials as potentially problematic. Construction of a tolerance band reduces such false alarms. The wider the tolerance band, the fewer false alarms, but at the cost of fewer truly problematic trials being captured.

A DiVBTA tolerance band can be determined using parametric or non-parametric methods based on the standard trial dynamics of a given clinical response variable (e.g., blood pressure or quality of life) and its statistical characteristics (e.g., absence of mean-variance relationships or ceiling effects).

A parametric approach to define tolerance bands which account for standard trial dynamics is to construct the 1-α DiVBTA prediction intervals based on a meta-regression [31]. Let $DiVBTA_{ijk}$ and $sd_{ijk}$ be the estimator and standard deviation for the $i^{th}$ variance difference or variance ratio between the control arm and treatment arm $i$, at the $j^{th}$ post-intervention time, and for the $k^{th}$ trial. A meta-analysis of these $DiVBTA_{ijk}$ leads to (1) the mean DiVBTA for the group of randomized trials ($M^*$), (2) the sample estimate of the variance of the true effect sizes ($T^2$), and (3) the variance ($V_{M^*}$) of the mean effect sizes, $M^*$ [32]. The prediction interval of the differences in variance between trial arms can be calculated as:

$$DiVBTA_{U,L} = M^* \pm t_{df}^{\alpha}\sqrt{T^2 + V_{M^*}}$$

where $t^{\alpha}$ is the t-value corresponding to the desired prediction interval (e.g., $\alpha \approx 0.0027$ for a 3-sigma probability). For independent DiVBTAs (i.e., one DiVBTA per clinical trial), the degrees of freedom (df) is 2 less than the number of clinical trials, and $V_{M^*}$ can be derived from a model-based estimate [32]. For correlated DiVBTAs (e.g., trials with more than 2 arms), robust meta-regression methods can be used where df is typically recommended to be calculated using a Satterthwaite approximation, and $V_{M^*}$ is estimated empirically using a cluster-robust "sandwich" estimator [33].

A non-parametric approach to define tolerance bands which account for standard trial dynamics is to derive the median DiVBTA for each included trial (median across all $DiVBTA_{ijk}$ for a given trial $k$). The median and the interquartile range of these median DiVBTAs can then be used as basis to construct DiVBTA Tukey inner and outer fences [34]. Potentially problematic trials can then be defined as statistically significant DiVBTAs which fall outside of these inner and outer Tukey fences.

The parametric or non-parametric cutoff values (e.g., $\alpha \approx 0.0027$ or inner Tukey fences) determine the width of the DiVBTA tolerance bands which in turn determine the sensitivity and specificity of the decision rule to identify potentially problematic trials. On one hand, defining a narrow tolerance band will increase sensitivity and decrease specificity; standard trial dynamics will frequently be falsely flagged as potentially problematic. On the other hand, setting a wide tolerance band will decrease sensitivity but increase specificity; potentially problematic trials will frequently fail to be flagged for further evaluation.

## Simulation methods

The simulation methods are presented using the ADEMP structure [35].

**Aims.** To evaluate the specificity and sensitivity of the proposed DiVBTA-based decision rule for detecting anomalous variance patterns under realistic and manipulated trial scenarios.

Specifically, the simulations assess:

- Specificity (true negative rate): the probability that the test correctly classifies non-problematic trials as non-problematic (i.e., avoids false positives/ false alarms) under standard/legitimate trial dynamics and non-standard/questionable trial dynamics, including (i) proper randomization, (ii) heterogeneous treatment effects (HTE), and (iii) missing-not-at-random (MNAR) dropout.

- Sensitivity (true positive rate): the probability that the test correctly identifies potentially problematic trials. For the simulations, a specific form of data manipulation (i.e., fraud) was modeled, namely deletion of a fraction of the worst

responders in the treatment arm followed by replacement (duplication) of those values with copies of the best-responder observations. How trial data is manipulated is largely unknown, making the relevance of this specific form of simulated fraud data questionable. As the mechanisms of fraud are largely unknown, the sensitivity of any method for identifying potentially problematic trials is difficult to quantify. Proposed decision rules will have utility if it has high specificity (i.e., few false positives), thus allowing it to be a screening tool to rule in potential fraud.

A secondary aim is to examine the performance of the proposed decision rule under both large-sample and small-sample scenarios. These aims focus on assessing robustness against false positive conclusions (specificity under realistic trial features that should not trigger alarms) and detection power (sensitivity to detect a targeted fraudulent mechanism), treating a statistically significant DiVBTA outlier as a diagnostic flag for potential issues (unintentional errors or fraud).

**Data-generating mechanisms.** The data-generating mechanisms for the simulations to assess these specific aims were based on five parameter estimates; two parameter estimates characterizing the probability distribution of the clinical response variable (e.g., the mean and the standard deviation of normal distribution) and three parameter estimates derived from a meta-analysis of clinical trials on the clinical response variable (mean ($M^*$), variance of the mean ($V_{M^*}$), and variance of the true effect size($\tau^2$)). Almost all clinical outcomes will be able to be modeled based on 5 parameter estimates, making the provided R-program versatile.

These 5 parameter estimates are specific to the (continuous) clinical response under investigation. Clinical outcomes such as patient-reported outcome measures can have ceiling or floor effects (which can create statistically significant DiVBTAs) whereas other clinical outcomes such as blood pressure do not suffer from this effect. Other standard trial dynamics such as the presence of heterogeneous treatment effects (which can create statistically significant DiVBTAs) can also differ depending on the selected clinical outcome. In other words, the 5 parameter estimates are specific to a specific response variable or domain.

The 5 parameters in the simulations presented here are diabetes-specific. Data were generated for simulated two-arm randomized trials with a continuous primary outcome (post-intervention HbA1c), modeled parametrically as a gamma distribution. Shape and rate parameters for the gamma distribution were estimated from individual participant data in a pivotal NIH-funded diabetes trial [36]. The standard trial dynamics in diabetes clinical trials were derived from a meta-analysis of post-intervention HbA1c standard deviations in 175 trials. The between-trial lnCVR heterogeneity was modeled as a baseline layer that existed in every simulated trial by sampling a trial-specific true lnCVR from $N(M^*, \tau^2)$.

Three standard (legitimate) trial dynamics were modeled to evaluate specificity:

- Randomization — Participants randomly assigned to treatment or control arms (assessed under the assumption of no heterogeneous treatment effects and no missing data to isolate randomization effects on lnCVR variability).

- Heterogeneous treatment effects (HTE) — Systematic variation in treatment response in the intervention arm, modeled via two parameters: (1) proportion of participants responding to treatment (varied 0%–50%), and (2) magnitude of treatment effect among responders (HbA1c treatment effect from −0.2% to −2.0%). Standard and non-standard treatment effect sizes were defined as between -0.2 to −1.4% and −1.4% to −2.0%, respectively. (−1.4% is the 3-sigma bound for the treatment effect sizes in a systematic review of diabetes trials; next section). A systematic review of diabetes trials failed to provide strong evidence in support of heterogeneous treatment effects. Standard and non-standard trial dynamics for the proportion of participants responding to treatment were defined as 0% to 20% and 30% to 50%, respectively.

- Missing-not-at-random (MNAR) dropout — Dropout probability dependent on unobserved (missing) outcomes modeled as deletion of the worst responders (highest HbA1c values) from the control group. This is an extreme mechanism unlikely in real settings and specified as such to stress-test specificity. Less than 3% of the trials included in Cochrane reviews have a dropout rate of 30% of more [37] and the fraction of these trials having the extreme mechanism of informative dropout modeled in this study is likely to very small. We classified the simulated dropout rate of 0% to 20% as a

standard trial dynamic, and a simulated dropout of 30% to 50% or larger (both with the extreme mechanism described above) as a non-standard trial dynamic. Given the extreme form of dropout modeled this is likely to a conservative definition of standard/non-standard.

One fraudulent mechanism was modeled to evaluate sensitivity:

- Deletion of a fraction of the worst responders (highest HbA1c) in the treatment arm, replaced by duplication of the single best responder (lowest post-treatment HbA1c) observation. The fraction replaced was varied between 50% and 90%.

Simulations were run in both large-sample (n = 250 per arm) and small-sample (n = 20 per arm) scenarios, reflecting approximate 95th and 25th quantiles of sample sizes in the motivating diabetes trial meta-analysis. For each scenario/combination, 1,000 independent trials (iterations) were simulated which leads to a Monte Carlo standard error for a sensitivity or specificity of 0.95 of ~0.007. The 3-sigma tolerance bounds for outlier classification were derived from a robust meta-analysis of 175 diabetes trials, specifically using the variance of true effects ($\tau^2$) and variance of the mean effect ($V_{M^*}$) estimated from lnCVR measures in those trials. All simulation code (R), random seeds, and modifiable key parameters are available at https://www.github.com/mhujoel/DIVBTA enabling verification and adaptation to other clinical settings.

**Estimand.** The estimand is the trial-specific true lnCVR. The estimator is the log coefficient of variation ratio (lnCVR), which quantifies relative variability between treatment and control arms. LnCVR is defined as: as

$ln\left(\dfrac{CV_T}{CV_C}\right) + \dfrac{1}{2}\left(\dfrac{1}{n_T-1} - \dfrac{1}{n_C-1}\right) + \dfrac{1}{2}\left(\dfrac{s_C^2}{n_C\,\bar{x}_C^2} - \dfrac{s_T^2}{n_T\,\bar{x}_T^2}\right).$ The approximate sampling variance of lnCVR is

$\dfrac{s_C^2}{n_C\,\bar{x}_C^2} + \dfrac{s_C^4}{2n_C^2\bar{x}_C^4} + \dfrac{n_C}{2(n_C-1)^2} + \dfrac{s_T^2}{n_T\,\bar{x}_T^2} + \dfrac{s_T^4}{2n_T^2\bar{x}_T^4} + \dfrac{n_T}{2(n_T-1)^2}.$

**Methods.** For each simulated trial (iteration), lnCVR and its sampling variance were computed using the formulas above. The decision rule to classify a trial as potentially problematic consisted of two criteria:

1. lnCVR is statistically significant ($p \leq 0.05$) using a two-sample t-test with Welch–Satterthwaite degrees of freedom for parallel-arm trials.

2. lnCVR exceeds the 3-sigma tolerance bound, derived from the meta-analysis of 175 diabetes trials defined as reliable (see next section).

No comparator methods were evaluated because comparators such as lnVR or F are biased since mean-variance relationships are present for the clinical outcome selected (Hemoglobin A1c).

**Performance measures.** The primary performance measures were specificity and sensitivity (framed in diagnostic testing terminology, where the "disease" is a problematic trial due to unintentional errors or fraud, and a positive test for "disease" is a statistically significant 3-sigma lnCVR outlier). These measures directly address the aims: high specificity indicates low risk of false alarms under legitimate trial features; high sensitivity indicates good detection of the targeted fraud mechanism.

## Methods for case-study assessing real-world utility

A systematic search was performed of the Cochrane library for systematic reviews with the key words of diabetes and glycaemic (("The Cochrane database of systematic reviews"[Journal]) AND (("2010/1/1"[Date – Publication]: "2023/05/23"[Date – Publication]))) AND (diabetes[Title] OR diabetic[Title]) AND (glycaemic OR "glucose-lowering drugs"). Key trial characteristics of the identified trials were abstracted and screened for the availability of post-intervention standard deviations or standard errors. Variance measures derived from statistical methods which assume a homogeneity of variance were excluded. The Carlisle-Stouffer-Fisher p-value, a measure of the plausibility of randomization given the baseline data, was calculated for trials reporting baseline data [15,38]. Risk of bias scores for randomization, blinding, and ascertainment were abstracted from the Cochrane reviews and assigned the values of 0 for high risk, 1 for unclear risk,

and 2 for low risk, respectively. Data source ("published data only" or relying on "published and unpublished data"), funding source, sample sizes, number of trials arms, trial duration, effect size, PubMed identification numbers (PMID) were abstracted. Trials without PMID included Ph.D. or Master's theses, grey literature, clinical trial registries which posted results on the clinical trial registration website, and other data sources not indexed in PubMed. The origin of the trial data was reclassified from "published data only" to "published and unpublished data" for 19 trials included in two Cochrane reviews because it was an organization other than Cochrane (the Institut für Qualität und Wirtschaftlichkeit im Gesundheitswesen or (IQWiG)) which obtained unpublished data for respectively 6 and 13 trials in two Cochrane reviews [39,40].

Trials with no or improbable baseline data were identified (operationalized as a one-sided Carlisle-Stouffer-Fisher p-value which was < 0.025, > 0.975, or missing) and were excluded from the set of trials for defining the tolerance band. Bootstrapping sampling assessed the robustness of excluding trials with no or improbable baseline data on the width of the tolerance band. Parametric and non-parametric tolerance bands were estimated as described in the previous section. Statistical tests reported in the trials were described as questionable when the trial failed to provide a description of the primary statistical test or reported any of the following tests without accommodation for the extreme variance heterogeneity (1) the use of standard Student's t-test (or equivalent pooled-variance method), (2) reliance on standard ANOVA (including repeated measures ANOVA), (3) use of standard repeated measures ANCOVA (or ANCOVA), (4) reporting of parametric or non-parametric tests "as appropriate" as this assumes readers can retroactively infer the decision rule, or (5) reporting of p-values without reporting the statistical tests used. Countries with a retraction rate above 0.10% in the field of medicine were labeled as having a high retraction ranking [41].

## Results

### Simulation results

The specificity and sensitivity of the proposed decision rule to flag potentially problematic trials are presented for standard and non-standard trial dynamics (Tables 1–3 for a 3-sigma decision rule, and S1–S3 Tables for a 4-sigma decision rule).

**Table 1. Specificity of 3-sigma statistically-significant lnCVR when randomization and heterogeneous treatment effects occur in legitimate trials, by sample size per trial arm.**

| HbA1c Effect size | Subgroup with HTE 10% | | Subgroup with HTE 20% | | Subgroup with HTE 30%* | | Subgroup with HTE 40%* | | Subgroup with HTE 50%* | |
|---|---|---|---|---|---|---|---|---|---|---|
| | n=20 | n=250 | n=20 | n=250 | n=20 | n=250 | n=20 | n=250 | n=20 | n=250 |
| −0.0% | 91.8% | 99.4% | 92.9% | 99.6% | 92.2% | 99.5% | 93.5% | 99.4% | 92.8% | 99.7% |
| −0.2% | 93.7% | 99.6% | 92.3% | 99.5% | 93.0% | 100.0% | 92.8% | 99.5% | 92.4% | 99.3% |
| −0.4% | 92.8% | 99.1% | 92.2% | 99.2% | 91.0% | 99.4% | 94.1% | 99.4% | 90.7% | 99.4% |
| −0.6% | 91.9% | 99.1% | 92.0% | 99.5% | 92.7% | 99.9% | 92.1% | 99.3% | 93.9% | 99.6% |
| −0.8% | 92.1% | 99.4% | 92.1% | 99.8% | 93.3% | 99.2% | 90.8% | 98.7% | 91.4% | 99.5% |
| −1.0% | 92.6% | 99.5% | 92.6% | 99.0% | 90.8% | 99.0% | 89.1% | 99.3% | 91.3% | 97.7% |
| −1.2% | 93.7% | 99.6% | 90.5% | 99.2% | 89.7% | 98.4% | 87.9% | 97.3% | 86.5% | 97.2% |
| −1.4%* | 91.2% | 99.1% | 90.7% | 98.5% | 88.3% | 98.2% | 84.9% | 96.6% | 82.3% | 94.3% |
| −1.6%* | 90.0% | 99.3% | 86.4% | 98.3% | 85.5% | 96.2% | 81.5% | 94.0% | 75.0% | 89.6% |
| −1.8%* | 90.6% | 99.4% | 85.8% | 96.9% | 80.6% | 94.1% | 73.7% | 89.0% | 69.4% | 81.5% |
| −2.0%* | 90.6% | 98.2% | 84.1% | 95.4% | 73.5% | 89.4% | 66.1% | 79.0% | 61.8% | 65.0% |

* The systematic review of diabetes trials included in this report suggests that treatment effects larger than a −1.2% and subgroup effects larger than 30% may not reflect standard trial dynamics. Specifically, the 3-sigma bounds of treatment effect sizes ranged from −1.4% to +0.8% and the meta-analysis of lnCVRs did not find convincing evidence of substantial heterogeneity of variances, suggesting that trial where large subgroups (e.g., 30%+) respond very differently in the intervention arm are unlikely.

**Table 2. Specificity of 3-sigma statistically significant lnCVR when randomization and an extreme form of MNAR dropout (0–50%) occur in trials, by sample size per trial arm.**

| Prevalence of missing-not-at-random dropout level | Sample size per clinical trial arm | |
|---|---|---|
| | n = 20 | n = 250 |
| 0 | 93.3% | 99.6% |
| 10% | 89.9% | 96.6% |
| 20% | 83.0% | 89.1% |
| 30%* | 77.3% | 79.0% |
| 40%* | 73.5% | 63.3% |
| 50%* | 69.0% | 51.5% |

* 30% and larger highly informative dropout rates are not a standard trial dynamic.

**Table 3. Sensitivity of 3-sigma statistically significant lnCVR for detecting simulated fraud (50–90% worst HbA1c scores in intervention arm replaced by best-responder value), by sample size per trial arm.**

| Proportion 'worst' responders deleted and replaced by "best" response. | Sample size per clinical trial arm | |
|---|---|---|
| | n = 20 | n = 250 |
| 50%* | 35.0% | 10.5% |
| 60%* | 49.0% | 8.7% |
| 70%* | 70.4% | 10.5% |
| 80%* | 89.8% | 34.6% |
| 90%* | 99.2% | 85.4% |

* Deleting 50% or more of the individual participant data cannot be considered subtle or minimal data fabrication. Stated differently, these simulation statistics evaluate the sensitivity to detect massive data fraud.

Specificity of 3- and 4-sigma significant lnCVRs in the presence of randomization only: Randomization alone leads to few false alarms. In large samples, the specificity of 3- and 4-sigma significant lnCVRs exceeds 99.4% and 99.9%, respectively (very few false alarms). In small samples, the specificity of 3- and 4-sigma lnCVRs exceeds 91.8% and 97.2%, respectively.

Specificity of 3- and 4-sigma significant lnCVRs in the presence of randomization and heterogeneous treatment effects: (i) Under standard trial dynamics (heterogeneous treatment effects combined with randomization), 3-sigma significant lnCVRs yielded 99.0% to 99.8% specificity in large-sample settings and 90.5% to 93.7% specificity in small-sample settings. By contrast, 4-sigma significant lnCVRs yielded 99.9% to 100% specificity in large-sample settings and 96.7% to 98.5% specificity in small-sample settings. (ii) Under non-standard trial dynamics (heterogeneous treatment effects combined with randomization), 3-sigma significant lnCVRs yielded 65.0% to 100% specificity in large-sample settings and 61.8% to 94.1% specificity in small-sample settings. By contrast, 4-sigma significant lnCVRs yielded 96.6% to 100% specificity in large-sample settings and 80.1% to 98.6% specificity in small-sample settings.

Specificity of 3- and 4-sigma significant lnCVRs in the presence of randomization and missing-not-at-random dropout: (i) Under standard-trial dynamics, when the prevalence of the extreme form of dropout modeled was 20% or less, the specificity exceeded 89.1% in large-sample settings and 83.0% in small-sample settings for 3-sigma significant lnCVRs. For 4-sigma significant lnCVRs, the specificity exceeded 98.2% and 90.9% in large- and small-sample settings, respectively. (ii) Under non-standard trial dynamics, for 3-sigma bounds, the specificity ranged from 69.0% to 77.3%

for small-sample settings, and from 51.5% to 79.0% for large-sample settings. Under non-standard trial dynamics, for 4-sigma bounds, the specificity ranged from 69.0% to 84.8% for small-sample settings, and from 82.5% to 95.4% for large-sample settings.

Sensitivity of 3- and 4-sigma significant lnCVRs to duplication of responses: The proposed decision rule to flag potentially problematic trials is not sensitive to detecting a 50% duplication of responses in a trial arm. It is only when the rate of data duplication in the intervention arm reaches 80% or higher that the sensitivity becomes larger than 89.8% for small-sample trials. In large-sample settings, a data duplication rate of 90% leads to a sensitivity of 85.4%. The sensitivity decreased when a 4-sigma significant lnCVR tolerance was selected for the decision rule.

**A case study assessing utility of the proposed decision rule**

The PRISMA flow diagram shows the systematic selection process which led to a sample of 305 trials, 226 of which with an ability to calculate lnCVR (S1 Fig). Trials reporting calculable lnCVRs (n = 226) (versus those where no lnCVRs can be calculated (n = 79)) were more likely not to report funding, to be of shorter duration, to have fewer authors, not to be indexed in PubMed, to have a smaller sample size, to have fewer trial arms, and to originate from a country with a high retraction ranking (S4 Table).

The 3-sigma prediction interval for lnCVR (i.e., the selected DiVBTA) was −0.54 to 0.47 for 175 trials for trials with plausible baseline data (175 trials;229 lnCVR estimates). Bootstrap sampling showed a modest impact of restricting the estimation of the tolerance bands to trials reporting plausible baseline data (S5 Table). Bootstrapping from the available 226 trials led to 3-sigma prediction interval where the lower bound of the 95% confidence interval ranged from −0.67 to −0.46 and the upper bound ranged from 0.39 to 0.65.

Nineteen trials reported statistically significant lnCVRs falling outside the parametric 3-sigma prediction interval of −0.54 to 0.47 (Fig 1). These trials, when compared to the 207 trials with either non-significant LnCVRs or LnCVRs not falling outside of the 3-sigma prediction interval, were more likely to report baseline data distributions which are inconsistent with randomization and smaller sample size (S6 Table). 18 of 19 trials reported at least one additional potentially problematic feature (Table 4): (1) improbable or no baseline data (n = 7), (2) calculation or data errors in glycemic responses (n = 3), (3) 0% dropout (n = 4), (4) high risk of attrition bias (n = 2) of allocation concealment bias (n = 2), (5) a larger than 3-sigma effect size for HbA1c improvement (n = 1), and (6) reporting of a questionable statistical test (e.g., standard Student t test), or no statistical test (n = 13). Eleven of the 19 trials reported statistically significant lnCVRs falling outside the parametric 4-sigma bounds.

The high prevalence of potentially problematic trials reported here (~8%) reflects on calendar years when awareness and scrutiny of problematic trials was low. Included Cochrane reviews were published starting in 2010; all but one of the Cochrane reviews included were published before Cochrane's 2021 editorial policy on managing potentially problematic studies. Authors of the most recent Cochrane review included in this report may have been unaware of the 2021 editorial policy, in part because it exists separately from the Cochrane Handbook.

**Discussion**

Three lines of evidence support the view that trials with unusually large and statistically significant Differences in Variances Between Trial Arms (DiVBTAs) can be flagged as potentially problematic. First, empirical evidence has shown that DiVBTAs are typically small and statistically insignificant [21,23–27,42], making trials with large and statistically significant DiVBTAs unexpected. Second, simulations demonstrated that the decision rule to flag studies with statistically significant DiVBTA outliers as potentially problematic has high specificity, especially when evaluated under standard trial dynamics. A high specificity implies that a flagged trial can reliably be ruled in as potentially problematic. Third, the real-world case study showed that the effort to calculate and analyze DiVBTAs is worthwhile. About 8% of the trials were flagged even when the definition of a large DiVBTA was defined as a 3-sigma event. These 3 lines of evidence suggest that identifying

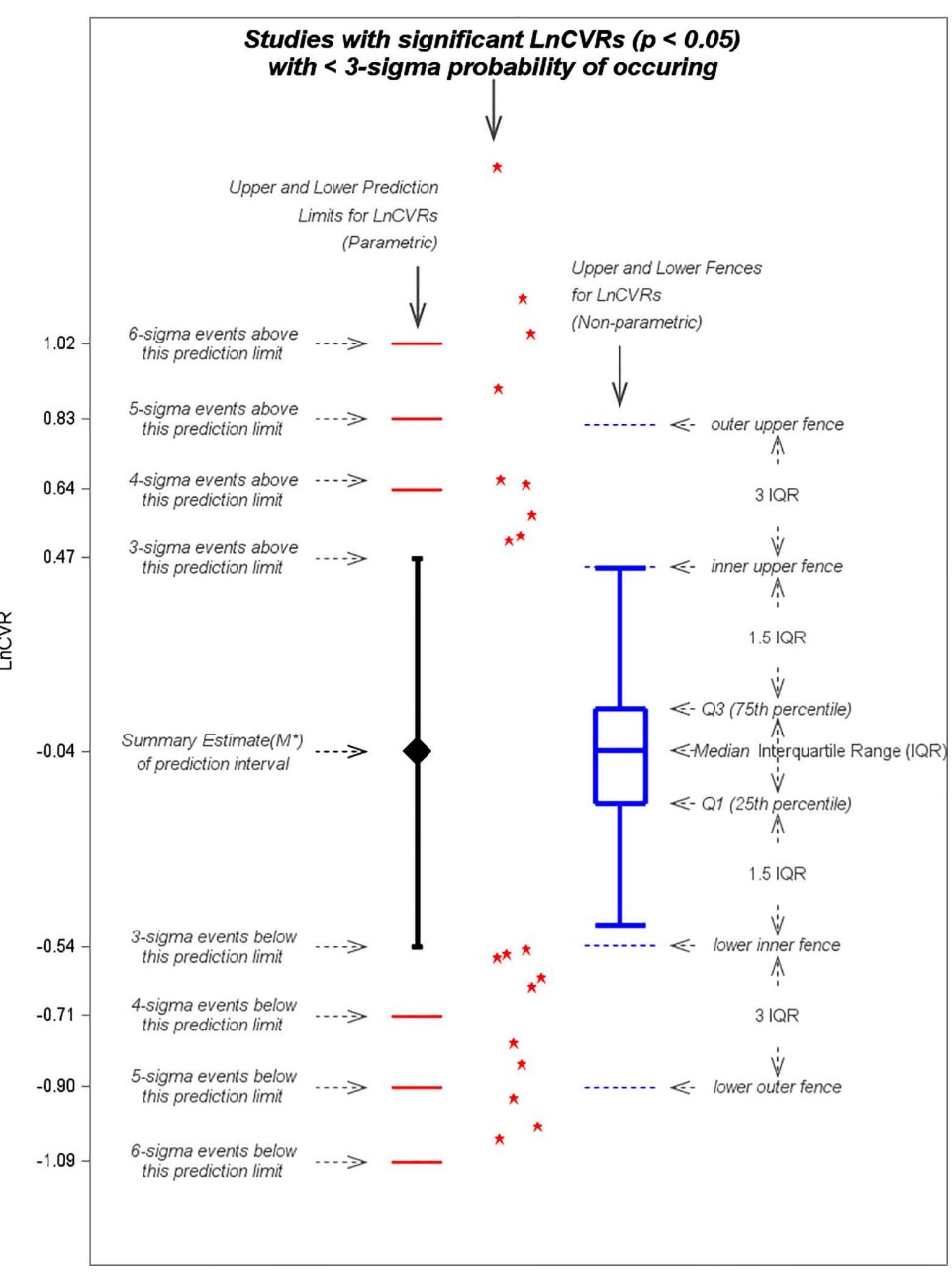

**Fig 1. Nineteen trials flagged as potentially problematic because their LnCVRs are (i) statistically significant and (ii) outside of the 3-sigma lnCVR prediction intervals estimated based on 175 trials.** The left side of the graph shows the parametric approach to screen for potentially problematic trials; construct the 3-sigma lnCVR prediction interval. The right side of the graph shows the non-parametric approach with the construction of Tukey inner and outer fences to screen for potentially problematic trials. The parametric 3-sigma prediction interval and the Tukey inner fences are remarkably similar.

**Table 4. Nineteen trials with DIVBTAs outside of the 3-sigma bounds – 7 checks on trustworthiness.**

| First Author | Ref. | Problematic or no baseline data[1] | Calculation/data error in glycemic respons[2] | Zero patients lost to follow-up[3] | High risk for allocation conceal-ment bias[4] | High risk for attrition bias[5] | Statistically significant HbA1c effect size> than 3-sigma[6] | Questionable statistical test[7] |
|---|---|---|---|---|---|---|---|---|
| Meschi | [46] | ✓ | | | | | | ✓ |
| Homko | [47] | | | | | | | ✓ |
| Vincent | [48] | | | | | | | ✓ |
| Kiran | [49] | | ✓ | ✓ | | | | |
| Macedo | [50] | | | ✓ | | | | |
| Agurs-Collins* | [51] | | | | | | | |
| Zieve* | [52] | | | | | ✓ | | |
| Durán* | [43] | | | | | | | ✓ |
| Tsalikis* | [53] | | | | ✓ | ✓ | | |
| Schiel* | [54] | ✓ | | | | | | ✓ |
| Huang* | [55] | ✓ | | ✓ | | | ✓ | ✓ |
| Hirsch* | [56] | | | | | | | ✓ |
| Schade* | [57] | ✓ | | | | | | |
| Ma | [58] | | ✓ | | ✓ | | | ✓ |
| Petrovski | [59] | ✓ | | | | | | ✓ |
| Al-Zahrani | [60] | | | | | | | ✓ |
| Fang | [61] | ✓ | | | | | | ✓ |
| Cohen | [62] | ✓ | | | | | | ✓ |
| Dans | [63] | | ✓ | ✓ | | | | ✓ |

* 8 trials with LnCVR sizes between the 3 and 4-sigma bounds. The checks for other problematic trial features in this table were in part derived from three currently suggested checklists to assess trustworthiness in randomized controlled trials [10–12] 1. Problematic baseline data: no baseline characteristics provided in the publication (leading to Carlisle-Stouffer-Fisher statistics which is missing), or a close to perfect balance for multiple baseline characteristics, or significant/large differences between baseline characteristics (Carlisle-Stouffer-Fisher p-value < 0.025, or > 0.975). 2. A (un)intentional data error or calculation error in the primary outcome (HbA1c): (a) standard deviations (before, after, and of the change) resulted in a correlation less than −1 or greater than +1, or (b) discrepancy in reported before, after, and HbA1c change. 3. Trial duration of 3 months or longer with Cochrane reported 0% dropout. 4. Cochrane reported a "High risk of bias for allocational concealment". 5. Cochrane reported a "High risk of attrition bias". 6 HbA1c effect size fell outside the 3-sigma bounds (−1.4% to 0.8%) and was statistically significant. 7. The reported statistical tests was unspecified or failed to specify that the heterogeneity of variances was addressed.

studies with statistically significant DiVBTA outliers is a worthwhile screening tool and offers a fair and objective flag for potentially problematic studies.

An illustrative example demonstrates how the proposed DiVBTA method provides an objective and fair alternative to the subjective impression of unusual variances that flagged problematic trials in the past. The 2009 Boldt et al. trial—pivotal in exposing one of the largest research misconduct scandals to date (22)—was originally questioned in part based on perceived anomalies in reported variances. Our method retrospectively confirms this concern objectively. The proposed methods here flagged the trial as potentially problematic. The DiVBTA was statistically significant ($p = 3.1 \times 10^{-13}$), a first criterion to flag a trial. The DiVBTA was also an outlier, the second criterion to flag a trial. This example illustrates that what was once flagged subjectively as potentially problematic can now be detected reliably and transparently using statistical criteria, thereby enhancing fairness and reproducibility in identifying potentially fraudulent or problematic studies. Further real-world examples of confirmed fraudulent trials (e.g., from retracted studies or audits) may reveal how often fraud manifests itself as outlying significant variance differences across trial arms.

It is essential to clarify that a statistically significant and unusually large DiVBTA identifies trials as potentially problematic but does not confirm fraud, fabrication, unintentional error, or any specific form of misconduct. Such outliers may occasionally arise from genuine, albeit rare, trial dynamics. As illustrated with the simulations with 3- and 4-sigma outliers as flags for identifying potentially problematic trials, the wider the tolerance band, the more unlikely a flagged trial is to have arisen from rare trial dynamics. DiVBTA outliers may also arise from unintentional errors beyond the control of the trial investigators (e.g., transcription mistakes during the publication process or in intermediary steps between trial publication by the clinical trial team and meta-analysis publication by a separate research team). In one such case, the published trial report presented only non-parametric statistics [43]. The trial investigators later provided unpublished parametric statistics directly to the meta-analysis team [44]—statistics now flagged as outliers in this report. When unpublished data are used in this way, it becomes impossible to determine whether an error occurred or, if it did, which team (trial investigators or meta-analyst team) was responsible.

These considerations underscore the role of DiVBTA as a screening tool rather than a definitive diagnostic, complementing rather than replacing other data integrity tools.

Spotting DIVBTA outliers could become integrated into checklists to assess the trustworthiness of clinical trial reports. Checks of baseline data such as the Carlisle-Stouffer-Fisher method focuses on detecting randomization failures [15]. DIVBTA methods extends scrutiny to post-randomization outcome data, capturing problematic data issues in post-baseline data, that may not be manifest at baseline. Its compatibility with summary statistics commonly reported in clinical trials makes it practical for implementation. The method's applicability to non-normal outcomes via lnCVR further enhances its versatility. By integrating parametric (robust meta-regression) and non-parametric (Tukey fences) approaches, and assessing different widths of tolerance bands, a fair and objective framework can be constructed to assess the robustness of labeling trials as potentially problematic. The case study presented suggests that trials with DiVBTA outliers frequently fail to report statistical tests which considers the extreme variance heterogeneity. This absence of an appropriate analysis can lead to biased p-values, risk invalid inferences and consequently expose patients to ineffective/harmful treatments or deny access to beneficial ones.

Despite its strengths, the DiVBTA method has limitations. First, detecting DiVBTA becomes challenging when reported standard deviations are imprecise. This can occur when small standard deviations (SD ≈ 1–2) are excessively rounded, an issue that becomes more pronounced when values are reported as standard errors, where rounding can result in a greater loss of information. Second, no meaningful DiVBTA can be computed for trials reporting standard deviations or standard errors calculated under the assumption of a homogeneity of variances. Third, simulation studies showed that small sample sizes increase the risk of false alarms (ruling in a trial as being potentially problematic, when in fact standard trial dynamics may have been responsible). Fourth, in the unlikely scenario in which there are no systematic reviews of clinical trials available, the proposed methods needs to start with a search for trials reporting variability estimates.

By offering a scalable, objective method, DiVBTAs could be integrated into the checklists of journal editors, peer reviewers, and meta-analysts to flag trials for further scrutiny, potentially reducing or preventing the impact of problematic studies on the meta-analyses which impact clinical guidelines. Several such workflows are already in place [13,45]. Prospective studies tracking flagged trials for retraction rates could quantify the method's predictive accuracy, while integration into automated tools or AI-assisted review systems could streamline its application. Collaborative efforts to combine DiVBTA with multimodal integrity checks (e.g., baseline anomalies, effect size outliers) could further improve specificity. Ultimately, the DiVBTA method may offer a robust, transparent approach to bolstering research integrity, addressing the urgent need for validated statistical tools to safeguard the credibility of clinical evidence.

## Supporting information

**S1 Text. DiVBTA measures other than lnCVR.**
(DOCX)

**S1 Table. Specificity of 4-sigma statistically significant lnCVR when randomization and heterogeneous treatment effects occur in legitimate trials, by sample size per trial arm.**
(DOCX)

**S2 Table. Specificity of 4-sigma statistically significant lnCVR when randomization and MNAR dropout (0–50%) occur in legitimate trials, by sample size per trial arm.**
(DOCX)

**S3 Table. Sensitivity of 4-sigma statistically significant lnCVR for detecting simulated fraud (50–90% worst HbA1c scores in intervention arm replaced by best-responder value), by sample size per trial arm.**
(DOCX)

**S1 Fig. Flow diagram of study selection for the meta-analysis of lnCVR estimates.** From 58 Cochrane reviews identified via search terms (n = 57) and follow-up (n = 1), 23 were excluded due to absence of HbA1c outcome or being reviews of reviews. 338 trials were identified in the remaining 35 Cochrane reviews, yielding 305 unique trials. 79 trials lacked informative SD estimates and were excluded, leaving 226 trials for the lnCVR meta-analysis.
(PNG)

**S4 Table. Characteristics of diabetes intervention trials stratified by reporting of outcome standard deviations.**
(DOCX)

**S5 Table. Characteristics of diabetes intervention trials stratified by reporting of statistically significant lnCVR outliers.**
(DOCX)

**S6 Table. Assessment of the robustness towards the selection of RCTs for inclusion into the lnCVR meta-analysis.** Bootstrap summary of lnCVR effect estimates and 3σ prediction interval bounds. Results from 1000 bootstrap replications (with replacement at the study level). The original values are based on the full dataset (n = 226 studies). The 95% confidence intervals are percentile-based.
(DOCX)

## Author contributions

**Conceptualization:** Philippe P. Hujoel.

**Data curation:** Philippe P. Hujoel.

**Formal analysis:** Philippe P. Hujoel, Margaux L.A. Hujoel.

**Methodology:** Philippe P. Hujoel, Margaux L.A. Hujoel.

**Supervision:** Margaux L.A. Hujoel.

**Writing – original draft:** Philippe P. Hujoel.

**Writing – review & editing:** Philippe P. Hujoel, Margaux L.A. Hujoel.

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
