## [Decision Letter · Decision Letter 0]

29 Jul 2025

Dear Dr. Hujoel,

Thank you for submitting your manuscript to PLOS ONE. After careful consideration, we feel that it has merit but does not fully meet PLOS ONE’s publication criteria as it currently stands. Therefore, we invite you to submit a revised version of the manuscript that addresses the points raised during the review process.

We look forward to receiving your revised manuscript.

Kind regards,

Robin Haunschild

Academic Editor

PLOS ONE

Journal Requirements:

2. Please note that your Data Availability Statement is currently missing the repository name and/or the DOI/accession number of each dataset OR a direct link to access each database. If your manuscript is accepted for publication, you will be asked to provide these details on a very short timeline. We therefore suggest that you provide this information now, though we will not hold up the peer review process if you are unable.

3. We note you have included a table to which you do not refer in the text of your manuscript. Please ensure that you refer to Table 3 in your text; if accepted, production will need this reference to link the reader to the Table.

Reviewers' comments:

Reviewer's Responses to Questions

**Comments to the Author**

1. Is the manuscript technically sound, and do the data support the conclusions?

Reviewer #1: Partly

Reviewer #2: Yes

2. Has the statistical analysis been performed appropriately and rigorously?

Reviewer #1: No

Reviewer #2: Yes

3. Have the authors made all data underlying the findings in their manuscript fully available?

Reviewer #1: No

Reviewer #2: No

4. Is the manuscript presented in an intelligible fashion and written in standard English?

Reviewer #1: Yes

Reviewer #2: Yes

Reviewer #1: The authors propose an estimator for implausible study outcomes which is a commendable initiative. The approach is beyond comparing means and less subjective. Nevertheless, the manuscript is of a very general nature. The applied setting is intuitive, however, the findings might be supported by a simulation study with all underlying assumptions being varied (treatment effects, dropout rates, the set of trustworthy trials, etc.).

Major

- page 4, 1st paragraph: respective DIVBTA estimators should be mentioned in more detail. The paper could also benefit from benchmarking results of alternative estimates in the applied setting

- page 4, 2nd paragraph: “The set of randomized controlled trials at the basis of clinical guidelines” and, in particular, the definition of a “set of trustworthy trials” should be extended. So far authors remain vague on this definition albeit it might have crucial relevance for application of DIVBTAs. Which criteria qualify to be part of the set, placebo-controlled trials, trials with active control (head-to-head comparisons), efficacy/safety trials, sample size of trials? When is the size of the set sufficient? How to flag a trial as untrustworthy in absence of such a set?

- The first assumption “First, there is no treatment effect heterogeneity – the treatment effect is the same for every individual” I'm not sure if this is mentioned here as a more general or specific assumption for the estimator? This cannot apply to placebo-controlled studies. If otherwise, please comment.

- Please specify in more detail the second assumption “Second, the error of the outcome does not depend on the treatment or the outcome”. How is error defined? Is that a plausible assumption for all kinds of outcomes? For example, single- or double-bounded outcomes should be considered, please see: https://doi.org/10.3102/1076998610396895. Variances should also change, in particular, when a considerable proportion of patients achieve remission.

- Please also be more detailed about randomization. The phrasing “A particularly unlucky randomization may lead to …” appears not well aligned with common terminology.

- The phrasing “the variability of DIVTBA can be impacted by the noise created by participant dropout” is unclear. In which form do dropouts add noise? To this reviewers experience, they introduce imbalanced patient characteristics and missingness. In case of selective dropout this leads to upwards/downwards biased results.

- The whole paragraph at the end of page 5/1st page 6 is unclear.

- The section “Interpretation of the DIVBTA variance observed in a set of trustworthy trials” might benefit from a table summarizing the assumptions for DIVBTA such as expectations regarding the mean/variability in trustworthy vs. untrustworthy trials, impact of dropout, interaction, by chance, trial size, randomization, and further biases.

Minor

Abstract: Please correct typos “. to illustrate this approach, we assessed the DIVBTAs for Hemoglobin A1cin a systematic sample”

Please revise the use and setting of references with/without leading spaces.

Reviewer #2: I very much enjoyed reading your manuscript and look forward to seeing it published soon.

With kind regards, Emma Sydenham (Senior Editor, Cochrane Database of Systematic Reviews)

Major comments:

1) Concerning point 3 above 'Have the authors made all data underlying the findings of their manuscript fully available' I have answered no as the list of reviews identified in 'II. Problematic differences in variance between trial arms (DIVBTA): Application' is not given. It would be helpful if the thirty-five Cochrane reviews that were identified and reported HbA1c could be referenced in this section, or listed in a table.

Minor comments:

1) The third sentence of the introduction does not follow. I suggest re-phrasing based on the following comment, and those below. While it is true that the unreliable trials have permeated clinical guidelines, I disagree that this has contributed to a growing crisis in research integrity. Rather, unreliable trials have permeated guidelines because research integrity and regulatory compliance criteria were not previously included in meta-analysis and guideline development processes. This is partly due to the information not being requested or made available by publishers in previous decades, and the issue not having been taken up by a working group in evidence based medicine years ago (i.e. in the early 2000s). I don't think there is a growing crisis in research integrity - I think research integrity was never raised as an issue until recently. Research integrity issues were highlighted during the COVID pandemic in particular, in 2020, which was some 30 years after evidence based medicine started taking hold as an academic discipline. I think the possibility of tackling research integrity issues in evidence based medicine has never been better than at present, because now there are a handful of research integrity checklists, policies, and methods (such as this paper) which are available for researchers to use which didn't previously exist.

Also, looking back ten years, the critics of research integrity were not morally wrong or factually incorrect in their views. (Here I refer to an old debate (https://www.bmj.com/content/350/bmj.h2463) and a more recent commentary (https://www.jclinepi.com/article/S0895-4356(25)00003-4/pdf).) Clinical trials are highly regulated and so it should indeed be safe to assume that they were conducted according to the relevant domestic and international regulations. It should be the case that study reports accurately reflect lawfully collected clinical trial data. One of a number of problems is that new computing technologies have been developed which make it easy to generate fake data, and these new technologies have become more accessible over the last ten years. Multiple issues relating to computer generated data, changes in medical journal publishing, trial registration, international harmonisation of clinical trial conduct and electronic data collection exist in parallel and are constantly evolving. The more recent acknowledgement that research integrity can be a problem, and the use of solutions, are making evidence based clinical guidelines safer. Here is one such example: https://www.cochrane.org/about-us/news/cochrane-launches-new-feature-identify-retracted-publications So I would encourage you to reconsider using the phrase 'a growing crisis in research integrity'.

2) The following sentence 'A 2021 Cochrane editorial...' could be expanded. This editorial by Boughton, Wilkinson and Bero was the announcement of Cochrane's Editorial Policy on Managing Potentially Problematic Studies, which was developed over 5 years with broad consultation. I assume I have been invited to comment on your work as I am listed as a member of the Policy advisory committee. Unfortunately the Policy document does not have its own DOI, because it is included in an online policy manual and website, so the editorial is often referenced instead of the policy implementation guidance web link. The editorial you reference was only one means of disseminating the policy, there are also training materials produced by Cochrane, there was a popular blog post by Richard Smith 'Time to assume that health research is fraudulent until proven otherwise?' https://blogs.bmj.com/bmj/2021/07/05/time-to-assume-that-health-research-is-fraudulent-until-proved-otherwise/ , among other resources and conference round table discussions.

However, rather than referring to prior calls for action, you could present the work as a contribution to the other new developments which have followed. For example, a research integrity tool with an associated R package was developed by Hunter et al: https://doi.org/10.1002%2Fjrsm.1738 and there is also an R package for the statistical checks of the REAPPRAISED checklist: https://reappraised.wordpress.com/2023/03/28/the-reappraised-r-package/ Ideally this piece of work will inform new R packages, which are currently being used to automate research integrity checks in the field of Data Science.

Are you aware that an R package has already been developed for this analysis? It might be worth referencing, too: https://github.com/harrietlmills/DetectingDifferencesInVariance

3) You have chosen to examine a cohort of Cochrane reviews for your example. The analysis that you have done for these trials is good, I just think there is a slight problem in the way you have explained the rationale for selecting these trials which should be reconsidered. The search for reviews starts in 2010 which is prior to the development of Cochrane's Editorial Policy for Managing Potentially Problematic Studies. So the trials that are included in your analysis are not ones which went through a Trustworthiness Screening Tool (such as: Identifying and handling potentially untrustworthy trials – Trustworthiness Screening Tool (TST). Developed by the Cochrane Pregnancy and Childbirth Group. Alfirevic Z, Kellie FJ, Weeks J, Stewart F, Jones L, Hampson L, on behalf of the Pregnancy and Childbirth Editorial Board and 10.1002/cesm.12037). The content of your work is fine and should be published, I just think the way you have framed the issue of the reviews being trustworthy because they are Cochrane reviews is not quite right if those reviews were published prior to Cochrane's policy and the reviews didn't incorporate a trustworthiness screening tool or another research integrity assessment tool or strategy.

4) Further to point 3) with regards to the paragraph 'Trustworthy trials as a source of DIVBTA estimates' (p.4), 'By focusing on a set of trials viewed as trustworthy by an authoritative organization' (p.6), and 'A non-parametric approach to define unusual DIVBTAs...' (p.7) it's worth noting that not all Cochrane authors are aware of the Policy on Managing Potentially Problematic Trials. This is partly due to the fact that the Cochrane Handbook is long, and the Policy is described in a separate online manual covering Cochrane's editorial policies. Even at present (July 2025), not every trial included in a Cochrane review is assessed using a research integrity tool. I have no doubt that will come in the future, but we aren't there yet and very few trials were statistically checked in the past. Furthermore, teaching about meta-analysis varies in scope and research integrity may not be included in the curriculum. Research integrity was not commonly taught prior to 2020, until the COVID pandemic brought issues concerning research integrity into public discourse.

5) I encourage you to re-phrase the sentences 'A non-parametric approach to define unusual DIVBTAs is to derive the median DIVBTA for each trustworthy trial' (p.7) and 'The two criteria for defining the set of trustworthy clinical trials were...' (p.10). I understand what you mean; however, I would like to point out that the trials you have selected for analysis have not been through a formal trustworthiness assessment prior to publication of the Cochrane reviews. The only Cochrane editorial group that routinely assessed all studies for trustworthiness prior to inclusion in the review was the Pregnancy and Childbirth Group, which developed and applied their Cochrane PCG-TST tool in the reviews for which they held editorial responsibility. (Note, this has all changed now as there is a 'new' Cochrane Central Editorial Service.) So while your work presented in this paper is true and valid in terms of the actual analysis, it is not the case that the studies included in the 35 Cochrane reviews were assessed as being trustworthy to start with. No trustworthiness assessment was done, apart from possibly filtering out the retracted studies as part of the searching procedures (as per the Mandatory standard C48 Examining Errata, Cochrane Handbook section 4.4.6 and the technical supplement section 3.9). The work you have done and presented in this paper is good and should be published, I'm just not sure you should say that the trials are trustworthy if they have not been through a trustworthiness screening tool or a research integrity checklist (such as PCG-TST, the Reappraised checklist (Grey et al), RIA (Weibel et al), TRACT (Mol et al), or a procedure as described in the RIGID Framework (Mousa et al)). These research integrity tools hadn't been developed in 2010 which is the start date of your search.

6) In the first paragraph of the Discussion, you could make reference to some other work in this area. For example, in order to understand the errors identified, the RIGID Framework and the Cochrane Policy recommend contacting trial authors to request clarification of the reasons for possible errors.

RIGID Framework: https://www.thelancet.com/pdfs/journals/eclinm/PIIS2589-5370(24)00296-7.pdf

Cochrane Policy: https://www.cochranelibrary.com/cdsr/editorial-policies/problematic-studies-implementation-guidance

Very sadly there have been a few cases internationally of researchers taking their own life after their work was found to have problems, and one of the reasons for liaising with them is to make them aware their work is under review and to give them an opportunity to explain any problems that might be identified. (You can look up the case of Yoshiki Sasai, for example, which was highly publicised. However, there are other cases which have received no publicity so the actual number of cases is slightly higher than one can find in a web search.)

7) Your example is in diabetes research, but you could also reference where a similar analysis has been used in other areas of medicine, such as: https://doi.org/10.1097/EDE.0000000000001401 and https://doi.org/10.1002/bimj.202200116 among others.

.

Reviewer #1: **Yes:**Dr. Adrian RichterDr. Adrian RichterDr. Adrian RichterDr. Adrian Richter

Reviewer #2: **Yes:**Emma Sydenham (Senior Editor, Cochrane Database of Systematic Reviews)Emma Sydenham (Senior Editor, Cochrane Database of Systematic Reviews)Emma Sydenham (Senior Editor, Cochrane Database of Systematic Reviews)Emma Sydenham (Senior Editor, Cochrane Database of Systematic Reviews)

---

## [Author Response · Author response to Decision Letter 1]

30 Nov 2025

Response to reviewers document was uploaded.

---

## [Decision Letter · Decision Letter 1]

13 Jan 2026

Dear Dr. Hujoel,

Thank you for submitting your manuscript to PLOS ONE. After careful consideration, we feel that it has merit but does not fully meet PLOS ONE’s publication criteria as it currently stands. Therefore, we invite you to submit a revised version of the manuscript that addresses the points raised during the review process.

We look forward to receiving your revised manuscript.

Kind regards,

Robin Haunschild

Academic Editor

PLOS One

Journal Requirements:

Reviewers' comments:

Reviewer's Responses to Questions

**Comments to the Author**

Reviewer #1: (No Response)

Reviewer #2: All comments have been addressed

2. Is the manuscript technically sound, and do the data support the conclusions?

Reviewer #1: Yes

Reviewer #2: Yes

3. Has the statistical analysis been performed appropriately and rigorously?

Reviewer #1: I Don't Know

Reviewer #2: Yes

4. Have the authors made all data underlying the findings in their manuscript fully available?

Reviewer #1: Yes

Reviewer #2: Yes

5. Is the manuscript presented in an intelligible fashion and written in standard English?

Reviewer #1: Yes

Reviewer #2: Yes

Reviewer #1: The authors have provided a very thorough and carefully considered revision, which has substantially improved the manuscript. The simulation studies convincingly underscore the relevance of an estimator that addresses implausible variance differences between trial arms and even provide novel insights in relation to previous methodological work, while also pointing to potentially problematic randomized controlled trials. The discussion of the proposed method is well balanced and appropriately acknowledges its limitations. It would therefore be highly desirable for this method to be considered in applied settings, for example in meta-analyses, to support the assessment of the underlying evidence base and its credibility, ideally via an openly available R package rather than a Git repository alone.

However, as the manuscript has undergone substantial revisions, there are remaining issues related to structure and naming conventions that at times make it difficult to follow. In particular, the manuscript appears to pursue four distinct and only loosely connected objectives, each addressed in a separate section:

(1) identification of statistically significant DiVBTA outliers,

(2) simulation studies,

(3) a case study on a landmark fraud case, and

(4) a case study on clinical trials included in systematic reviews on diabetes management.

As a consequence, the manuscript deviates from the conventional IMRaD structure. For example, methods are introduced in both Sections 1 and 2, and the latter simultaneously presents results. Moreover, the two sections appear somewhat disconnected, as estimates or methods introduced in Section 1 do not clearly reappear in Section 2. In this context, closer adherence to established reporting guidelines would substantially strengthen the manuscript. In particular, the recommendations for simulation studies proposed by Boulesteix et al. (1) and Morris et al. (2) (ADEMP framework, endorsed by the STRATOS initiative) would provide a helpful structure. At present, key elements required for transparent reporting of simulation studies are missing or insufficiently described, including:

- Why are results presented only for LnCVRs?

- How was the simulation setup defined with respect to the number of iterations, software packages used, distributional assumptions, and used seeds?

- How were noise levels and signal-to-noise ratios handled?

- Why was n = 20 chosen for small-sample trials, and is n = 250 a realistic or appropriate choice for large trials?

- Is the chosen range of treatment effect variability (0.2% to 1.4%) plausible for small randomized controlled trials?

In addition, clear performance metrics such as sensitivity and specificity are currently missing. Potential users of the proposed method need to understand both the risk of false-positive findings and the probability that an untrustworthy trial remains undetected.

Minor comments

- Inconsistent naming conventions are used, for example: “Large samples (250/trial arm) and HTE” versus “Small samples (n = 20 per group) and HTE”.

- The correct reference to Tukey’s original work should be included and how fences or spread are defined should be added.

(1) Boulesteix A-L, Groenwold RH, Abrahamowicz M, Binder H, Briel M, Hornung R, Morris TP, Rahnenführer J, Sauerbrei W. Introduction to statistical simulations in health research. BMJ Open. 2020;10(12):e039921. https://doi.org/10.1136/bmjopen-2020-039921.

(2) Morris TP, White IR, Crowther MJ. Using simulation studies to evaluate statistical methods. Statistics in Medicine. 2019;38(11):2074-102. https://doi.org/10.1002/sim.8086.

Reviewer #2: (No Response)

.

Reviewer #1: **Yes:**Dr. Adrian RichterDr. Adrian RichterDr. Adrian RichterDr. Adrian Richter

Reviewer #2: **Yes:**Emma SydenhamEmma SydenhamEmma SydenhamEmma Sydenham

---

## [Author Response · Author response to Decision Letter 2]

10 Mar 2026

Response to reviewer was uploaded as a word document

---

## [Editor Report · Decision Letter 2]

17 Mar 2026

Unusual Outcome Variances as a Method to Identify Potentially Problematic Clinical Trials

PONE-D-25-25920R2

Dear Dr. Hujoel,

We’re pleased to inform you that your manuscript has been judged scientifically suitable for publication and will be formally accepted for publication once it meets all outstanding technical requirements.

Kind regards,

Robin Haunschild

Academic Editor

PLOS One
---

## [Editor Report · Acceptance letter]

PONE-D-25-25920R2

PLOS One

Dear Dr. Hujoel,

I'm pleased to inform you that your manuscript has been deemed suitable for publication in PLOS One. Congratulations! Your manuscript is now being handed over to our production team.

Kind regards,

on behalf of

Dr. Robin Haunschild

Academic Editor

PLOS One